# Varicella outbreaks in schools and kindergartens in Shanghai, China from 2011 to 2020

**Jing Wang, Zhenhui Xu, Qiang Gao** [ORCID] *

Department of Immunization, Huangpu District Center for Disease Control and Prevention, Shanghai, China

* hpcdcpolio@126.com

## Abstract

### Background

Varicella is a contagious disease caused by varicella-zoster virus and varicella vaccine (VarV) is the most effective way to prevent and control varicella. Despite high VarV coverage there were still varicella outbreaks in schools and kindergartens. We aim to analyze the epidemiological characteristics of varicella outbreaks in Huangpu District, Shanghai, China from 2011 to 2020.

### Methods

For varicella outbreaks, case information and vaccination history were collected. Mann–Kendall test and descriptive methods were used to analyzed the trend and epidemiological catachrestic of varicella outbreaks.

### Results

A total of 57 varicella outbreaks were reported from 2011 to 2020, including 30 outbreaks (52.6%) in primary schools. The results of the Mann–Kendall trend test ($z = 1.97$, $p = 0.049$) showed an upward trend in the number of cases during the study period, but the trend change was not statistically significant. Emergency vaccination was carried out in 42 (73.7%) outbreaks which influenced the duration of the epidemic ($F = 4.53$, $p = 0.0379$). A total of 573 varicella cases were reported, including 357 cases (62.3%) who had received at least one dose of varicella vaccine.

### Conclusions

The number of varicella outbreaks has not changed significantly in the decade from 2011 to 2020. The strategy of varicella vaccination, the development and application of varicella vaccine, and the control measures after the occurrence of varicella outbreaks need to be optimized. In addition to vaccination, as a disease transmitted by contact, quarantine measures, good personal hygiene, environmental disinfection, and ventilation are also important.

**Data Availability Statement:** There is legal restrictions on sharing data publicly. But, researchers could request access to the data through email. The email address is Hpcdckjk@126.com. It is the email address of Mr.

Zhao Jiakui, who is the member of ethics committee of Huangpu CDC and also in charge of scientific research and teaching work in Huangpu CDC. The datasets analysed during the current study are not publicly available due to the provision on confidentiality of information of Huangpu CDC but are available upon request to Mr. Zhao Jiakui (Hpcdckjk@126.com) or the corresponding author. All data were anonymised before access.

**Funding:** This work was supported by Yangtze River Delta Regional Leading Talents Research Project on Immunization (CSJP032). The funders had no role in study design, data collection and analysis, decision to publish, or preparation of the manuscript.

**Competing interests:** The authors have declared that no competing interests exist.

## Introduction

Varicella is a contagious disease caused by varicella-zoster virus (VZV). It is usually a childhood infection, with the majority of cases occurring in people aged under 6 years [1]. Although the disease manifests as a mild skin rash, complications such as encephalitis, pneumonitis, and secondary bacterial infections can occur, resulting in hospitalization and/or death [2]. Varicella is a vaccine-preventable disease and the World Health Organization advises routine childhood immunization in countries with a significant public health impact of the disease [3]. Varicella vaccine (VarV) is the most effective way to prevent and control varicella [4]. In China, VarV was first used in 1998, and the current vaccination procedure is to administer one dose of VarV to children aged above 12 months [5]. For effective control of varicella outbreaks, Shanghai began to implement emergency vaccination with VarV at schools and kindergartens in 2014 [6]. Subsequently, a vaccination strategy involving two free doses of VarV was implemented in Shanghai in 2018. For this vaccination procedure, the two free doses of VarV are respectively administered at 12–28 months of age and 4 years of age. However, the interval between the two doses of VarV and the protective effect of the vaccine remain under debate, and breakthrough varicella cases have been found in varicella outbreaks [5]. Therefore, in the present study, we analyzed the epidemiological characteristics of varicella outbreaks in Shanghai from 2011 to 2020, with the aim of providing evidence for the effect of the VarV vaccination strategy and improving the vaccination procedure as well as the control and prevention measures for varicella outbreaks.

## Materials and methods

### Data collection

There were 68 schools with 50,125 students and 67 kindergartens with 12,127 children in Huangpu District, Shanghai, China in 2020. An outbreak of varicella was defined as an event in which more than five cases of varicella occurred within 21 days in the same kindergarten or school were reported. For each outbreak, information of varicella cases, emergence vaccination and other control and prevention measures are collected and reported to local CDC. Then a report about the outbreak is required written by CDC in accordance with the varicella control and prevention regulations in Shanghai. The time of each outbreak occurrence is defined as the onset time of the first case, and each school year is divided into spring and autumn semesters, comprising March to July and September to January of the following year, respectively. For the present study, investigation reports of varicella outbreaks from 2011 to 2020 were collected. Vaccination history was obtained by checking the immunization certificates of cases or the local immunization information system records. Breakthrough varicella cases were defined as cases of varicella occurring at >42 days after VarV immunization. For the analysis, those who were vaccinated within 42 days before varicella onset, those with unknown vaccination history, and those with previous varicella history were excluded.

This study was approved by the Ethics Review Board of Huangpu District Center for Disease Control and Prevention (No. 2021HPLL10). All data were anonymized before access. All methods were performed in accordance with the relevant guidelines and regulations. Consent to participate is not applicable.

### Statistical methods

Data on time of occurrence, case distribution, outbreak duration, and prevention and control measures undertaken were collected to determine the epidemic characteristics of varicella outbreaks in the past 10 years. Demographic characteristics and vaccination history were collected

to analyze the epidemiological characteristics of breakthrough varicella cases. All cases were anonymous, and no ethics were involved. Excel was used for data extraction and collection. The incidence risk was calculated to describe the occurrence of VZV in schools and kindergartens using the following equation: incidence risk = number of VZV cases / total students at schools or kindergartens.

The cases were summarized and analyzed according to the time of onset, and the characteristics of the varicella cases and outbreaks during 2011–2020 were described. The F test was employed to compare the differences in VZV cases among various factors using Stata 16.0 software.

The Mann–Kendall trend test was employed to evaluate the increase or decrease in the number of VZV cases during the study period using R 4.1.3 software. In the results, a z-value of $<0$ indicated a downward trend for the time series, while a z-value $>0$ indicated an upward trend for the time series. Furthermore, a p-value of $<0.01$ indicated that the trend change was statistically significant.

## Results

### Varicella outbreaks

A total of 57 varicella outbreaks were reported from 2011 to 2020, of which 15 (26.3%) occurred in 2016 and 12 (21.1%) occurred in 2018. There were 18 (31.6%) outbreaks in the spring semester and 37 (68.4%) in the autumn semester. Most of the outbreaks occurred in primary schools (30 outbreaks, 52.6%), high schools (14 outbreaks, 24.6%), and middle schools (9 outbreaks, 15.8%), with only 4 (7.0%) outbreaks occurring in kindergartens. A total of 573 cases of varicella were reported, including 47 cases in a single outbreak in 2019, within 14 classes. The longest outbreak duration was 90 days, the shortest was 3 days, and the average was 26.6 days (Table 1).

Based on the 21-day incubation period for varicella, 31 (54.4%) outbreaks lasted for one cycle, 20 (35.2%) outbreaks lasted for two cycles, and 6 (10.5%) outbreaks lasted for more than three cycles. In terms of epidemic scale, 34 (59.6%) outbreaks had a total number of cases below 10, 20 (35.1%) outbreaks had a total number of cases between 10 and 20, and 3 (5.3%) outbreaks had a total number of cases above 30. As the epidemic period became longer, the number of cases increased (F = 47.46, p<0.05). There were also some outbreaks that lasted for a long time, but had few cases (Fig 1).

As mentioned above, a total of 573 varicella cases were reported from 2011 to 2020. The results of the Mann–Kendall trend test (z = 1.97, p = 0.049) showed an upward trend in the number of cases during the study period, but the trend change was not statistically significant. The incidence risk increased slightly in 2014 after the strategy for free emergency vaccination was introduced in Shanghai. At the beginning of 2020, Shanghai implemented the policy of

**Table 1. Summary of varicella outbreaks in Huangpu District, Shanghai, China from 2011 to 2020.**

| Classification | No. of outbreaks | Mean | Std Error | Minimum | Maximum |
|---|---|---|---|---|---|
| No. of students | 57 | 752.1 | 390.8 | 124 | 2520 |
| No. of varicella cases | 57 | 10.1 | 6.7 | 5 | 47 |
| No. of classes with varicella cases | 57 | 3.3 | 2.2 | 1 | 14 |
| Interval between first and second cases (d) | 57 | 7.9 | 6.5 | 0 | 19 |
| Duration of outbreak (d) | 57 | 26.6 | 18.2 | 3 | 90 |
| No. of emergency vaccinations | 41 | 86.2 | 85.9 | 11 | 370 |
| Interval between first case and emergency vaccination (d) | 42 | 11.8 | 7.5 | 2 | 32 |
| Interval between second case and emergency vaccination (d) | 42 | 3.6 | 3.4 | -3 | 15 |

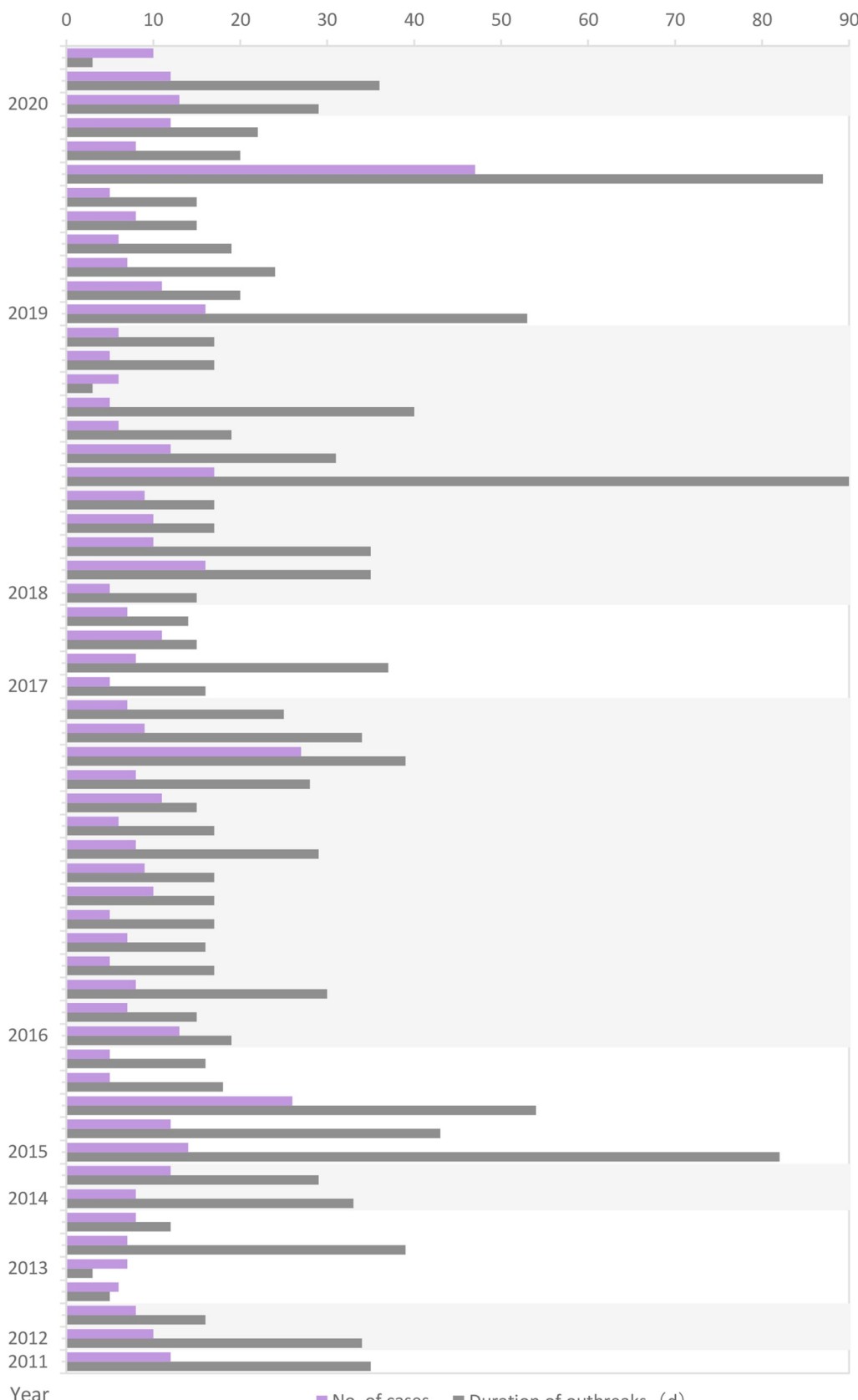

**Fig 1. Scale and duration of varicella outbreaks in Huangpu District, Shanghai, China from 2011–2020.**

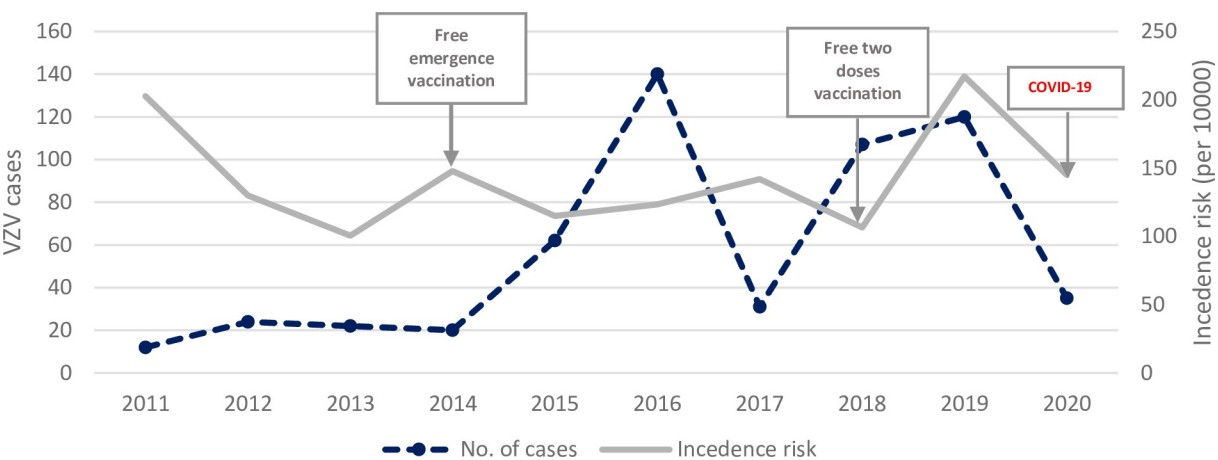

**Fig 2. VZV cases and incidence risk in Huangpu District, Shanghai, China from 2011 to 2020.**

suspending classes in primary and secondary schools and kindergartens because of the COVID-19 pandemic, and the number of reported cases dropped sharply. Meanwhile, the incidence risk did not change significantly (Fig 2).

## Emergency vaccination

Emergency vaccination was carried out in 42 (73.7%) outbreaks from 2011 to 2020. Emergency vaccination had no effect on the total number of reported cases (F = 3.72, p = 0.059), but was able to shorten the duration of the outbreak to some distance (F = 4.53, p = 0.0379).

In 12 outbreaks (28.6%), emergency vaccination was completed at 2 days after the onset of the secondary reported varicella case. The earliest emergency vaccination was carried out at 3 days before onset of the secondary case, and the latest was at 15 days after onset of the secondary case. The interval between onset of the secondary case and emergency vaccination was correlated with the scale of the outbreak and was statistically significant (F = 8.64, p = 0.0054). The earlier the emergency vaccination was carried out, the fewer cases of varicella there were.

## Breakthrough varicella cases

Among the total of 573 cases reported in the 57 outbreaks, 357 (62.3%) had a history of VarV vaccination before onset of the disease. Among these breakthrough varicella cases, 345 (96.6%) had a history of one dose of VarV and 12 (3.4%) had a history of two doses of VarV, and the cases comprised 227 (63.6%) male students and 130 (36.4%) female students. The mean interval was 9.0 years, the shortest interval was 0.5 years, and the longest interval was 16.4 years.

The clinical symptoms of breakthrough varicella cases included fever ($\geq 37.3°C$) in 85 (23.8%) cases, with average maximum body temperature of 37.9°C and highest body temperature of 40.0°C. Regarding other symptoms, 315 (88.2%) had a rash, 295 (82.6%) had vesicles, and 1 had a complication of encephalitis. The relationships between interval from last VarV dose to onset of varicella and clinical symptoms are shown in Table 2. Time interval from last dose of VarV vaccination to disease onset was related to occurrence of rash and vesicles, with statistical significance.

## Discussion

The United States proposed the implementation of emergency vaccination in 1999 [7] and a two-dose VarV vaccination strategy in 2007 [8]. By contrast free emergency vaccination with

**Table 2. Clinical symptoms of breakthrough varicella cases.**

| Symptoms | | No. of cases | Mean | Std Error | 95%IC | F | p |
|---|---|---|---|---|---|---|---|
| Fever | Yes | 85 | 9.7 | 4.2 | 8.8–10.6 | 3.80 | 0.052 |
| | No | 272 | 8.8 | 3.6 | 8.4–9.2 | | |
| Rash | Yes | 315 | 9.3 | 3.9 | 8.9–9.7 | 14.36 | 0.000 |
| | No | 42 | 7.0 | 1.6 | 6.5–7.5 | | |
| Vesicles | Yes | 295 | 9.3 | 4.0 | 8.8–9.7 | 6.56 | 0.011 |
| | No | 62 | 7.9 | 2.4 | 7.3–8.5 | | |

VarV was implemented in 2014 and VarV was included in the local immunization program with two free doses vaccination strategy in 2018 in Shanghai, China. It can be seen from the results of the present study that the number of varicella outbreaks did not change significantly from 2011 to 2020. Although there were few varicella outbreaks before 2014, the incidence risk was very high, indicating that the outbreaks were not effectively controlled. The incidence risk dropped after the strategy for free emergency vaccination was introduced in 2014. However, the emergency vaccination strategy did not seem to have a significant effect on the number of varicella outbreaks. On the contrary, the number of varicella outbreaks and cases increased significantly after 2015, and reached a peak in 2016 (15 outbreaks, 140 cases, incidence risk: 123.4 per 10000). Subsequently, the number of varicella cases and the incidence rate again reached high levels in 2019 (12 outbreaks, 107 cases, incidence risk: 106.6 per 10000) after the inclusion of VarV into the local immunization program in 2018. One varicella outbreak in a high school in 2019 had 47 reported cases and lasted for 87 days. Despite emergency vaccination of the entire school, the epidemic was not well controlled because of a delay in the emergency vaccination. Another reason is that, although the strategy for VarV vaccination is constantly being optimized in Shanghai, the impact of any change in strategy has a time delay. However, as can be seen from the results, emergency vaccination had some effect on the control of varicella outbreaks. Nevertheless, it is difficult to conclude whether this occurred because the schools that carried out emergency vaccination paid more attention to varicella outbreaks and took some other control and prevention measures. The effectiveness of the emergency vaccination strategy and the varicella policy requires further evaluation.

Vaccination of susceptible people in close contact with varicella cases can effectively prevent or slow down the occurrence of an epidemic [4]. We found that emergency vaccination was implemented in 42 (73.7%) outbreaks, and was able to shorten the duration of the outbreaks. These findings demonstrated that emergency vaccination with VarV had a positive effect on the control of varicella outbreaks in Shanghai. Based on the Shanghai varicella epidemic treatment plan, general emergency vaccination is required to be completed within 3 days after the occurrence of a varicella outbreak. The present results further revealed a correlation between the time of emergency vaccination and the scale of varicella outbreaks. The earlier emergency vaccination is carried out, the better the control of the outbreak is. The World Health Organization recommends that emergency vaccination should be carried out at 3–5 days after the outbreak, at which time the protection ratio can reach 90% [9]. However, considering medical staff availability, sufficient vaccine supply, and cold-chain equipment capacity, implementation of VarV emergency vaccination should be administered based on the specific situation of outbreaks to avoid blindly expanding the number and scope of the emergency vaccination, and consequently wasting resources.

In Shanghai, the VarV vaccination procedure until 2018 was one dose administered to children aged above 12 months. However, the results of the present study demonstrated that the protective effect of this one-dose vaccination strategy was poor, given that 62.3% of varicella

cases had a history of VarV vaccination before onset of illness. The protective effect of the vaccine diminishes over time, eventually leading to breakthrough varicella cases. Studies in the United States found that varicella outbreaks and breakthrough varicella cases still occurred after one dose of VarV vaccination, and indicated that even when the VarV vaccination coverage was high, the protective effect was close to 85%, which was not sufficient to prevent varicella outbreaks in schools and kindergartens [10, 11]. Therefore, it is necessary to change the vaccination strategy for VarV in China. In the United States, the one-dose VarV vaccination procedure started in 1996, and the two-dose VarV vaccination procedure started in 2006. The first dose was administered at 12–15 months of age, and the second dose was administered at 4–6 years of age [12]. The protective effect of the two-dose VarV strategy reached 94%-98% in the United States [10]. A study by Kauffmann and colleagues further confirmed that a two-dose VarV strategy was more effective than a one-dose VarV strategy in Germany [13].

Among the breakthrough varicella cases, mild clinical symptoms were found, with a low proportion of fever cases (23.8%), but a few cases with rashes and vesicles still occurred. Although the protective effect of VarV was not sufficient to prevent the occurrence of varicella, it was still able to alleviate the clinical symptoms to a certain extent. Furthermore, even though the clinical symptoms of varicella are mild, complications can occur in a very small number of cases, and the probability of complications increases with aging [14]. We also found one varicella case with a complication in the present study. The vaccination coverage of one-dose VarV vaccination in children aged 24 months was 91.8% in Shanghai in 2018 [6] and ≥90% in the United States from 2017 to 2019 [15]. It is necessary to maintain a high level of VarV vaccination coverage to prevent varicella outbreaks and reduce the risk of exposure to VZV [16].

Regarding the age for initializing VarV vaccination, a previous study found that an early age for the initial dose of VarV was a risk factor for breakthrough varicella cases [17], which may be related to the immature immune system and the gradually weakening antibody level in young children. Furthermore, the second dose of VarV should be administered at 3 or 5 years after the first dose to achieve a better protective effect [18]. We found that the current VarV vaccination strategy was not very effective in controlling outbreaks of varicella, and that the proportion of breakthrough varicella cases was high. Therefore, in addition to strengthening vaccination to improve the protective effect of VarV, it is necessary to introduce or develop new vaccines with better effects. A combination vaccine for mumps, rubella, and varicella vaccine (MMRV) has been used in the United States since 2008 [3], and its usage has reduced the number of vaccine doses and ensured vaccination coverage. At present, China has not introduced the MMRV. Therefore, we should pay attention to not only vaccine research and development, but also the introduction of new vaccines.

## Limitations

The present study has several notable limitations. The impact of the change in the VarV vaccination strategy on varicella outbreaks still requires follow-up research. Meanwhile, COVID-19 has not only caused school closures, but also changed the behavior patterns of students and teachers, and thus the impact of COVID-19 warrants further evaluation. Finally, the data for the study came from Shanghai, China, where VarV is included in the local immunization program, and therefore the findings have only limited generalizability for some other regions in China.

## Conclusions

It is worth noting that due to the control policy for COVID-19, all schools and kindergartens were closed from January 2020 to May 2020, and all students studied at home through online

classes. Under this policy, the numbers of varicella outbreaks and varicella cases were reduced in 2020 compared with other years. This means that quarantine measures remain the most effective measure to control varicella [19]. Specifically, there is no transmission of the disease without contact. Therefore, it is very important to isolated the infected class from other classes after an outbreak of varicella, and even suspend the class and stop teaching activities when necessary, to effectively control the spread of varicella in a timely manner. However, the cost and impact of doing so are high and decisions need to be made based on the specific circumstances. Furthermore, as a disease transmitted by contact, good personal hygiene, environmental disinfection, and ventilation are also important. Varicella outbreaks in school can be controlled through strict isolation combined with other interventions.

## Acknowledgments

The authors would like to thank staff of vaccination clinics in Huangpu District, who facilitated data collection.

## Author Contributions

**Conceptualization:** Jing Wang, Qiang Gao.

**Data curation:** Zhenhui Xu.

**Formal analysis:** Jing Wang, Zhenhui Xu.

**Writing – original draft:** Jing Wang, Qiang Gao.

**Writing – review & editing:** Zhenhui Xu, Qiang Gao.

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
