## [Decision Letter · Decision Letter 0]

11 Mar 2022

PONE-D-21-36096Varicella outbreaks in schools and kindergartens in Shanghai, China from 2011 to 2020PLOS ONE

Dear Dr. Gao,

Thank you for submitting your manuscript to PLOS ONE. Even after several attempts, we were able to find just one expert to review this manuscript. The reviewer raised some important issues to address. I have also some comments appended below. Therefore, we invite you to submit a revised version of the manuscript that addresses the points raised during the review process.

ACADEMIC EDITOR: I suggest using a statistical trend analysis (e.g., Mann–Kendall trend) to evaluate the increase or decrease in the number of cases over the study period.

I observed some values of F-statistic and P in different places in the result section. However, the use of different statistical tests to generate these values is missing in the statistical analysis section. Please incorporate.

Please describe the method to calculate the incidence risk. So far I understand the incidence rate [number of new cases divided by the total person-time at risk] was not calculated and so please replace incidence rate by incidence risk in Figure 2. Moreover, please replace the incidence rate (1/1000) with the incidence risk (per 10000) in the right vertical axis of figure 2.

We look forward to receiving your revised manuscript.

Kind regards,

A. K. M. Anisur Rahman, Ph.D.

Academic Editor

PLOS ONE

Journal Requirements: 

4. In your ethics statement in the manuscript and in the online submission form, please provide additional information about the patient records used in your retrospective study. Specifically, please ensure that you have discussed whether all data were fully anonymized before you accessed them, or whether the authors had any access to identifying patient information.

Reviewers' comments:

Reviewer's Responses to Questions

**Comments to the Author**

1. Is the manuscript technically sound, and do the data support the conclusions?

Reviewer #1: Yes

2. Has the statistical analysis been performed appropriately and rigorously? 

Reviewer #1: I Don't Know

3. Have the authors made all data underlying the findings in their manuscript fully available?

Reviewer #1: Yes

4. Is the manuscript presented in an intelligible fashion and written in standard English?

Reviewer #1: Yes

5. Review Comments to the Author

Reviewer #1: Wang et al presented the number of varicella outbreaks from 2011 to 2020 in China.

I have comments and suggestions:

• in your background would add more information about Varicella symptoms since you talked about it in your discussion.

• the number of varicella outbreaks has not changed significantly in the decade from 2011 to 2020 >>> at least in the abstract, no inclusion of trend data (instead supplied only average data).

• what do you mean in line 72-73 when you said case info is collected and a report was written? was it done by you or an official source?

• can you explain what do you mean when you mentioned in line 122 on the second day?

• line 132 it would be nice if you change the title from BV cases to breakthrough Varicella cases

• from lines 148 to 152 it is not clear what are you trying to say. the sentences are not connected.

• why does China only give one dose (compared to standard of practice in US of 2 doses)? Does this discrepancy drive outbreak?

• Clarify whether 2nd dose five if patients already received a dose prior to the outbreak

• How did they diagnose/define Varicella? Are there any clinical basis or labs?

• Of course, there are more cases in kindergarten since fewer students enrolled in school

• Was there an explanation of statistical analysis of trend?

• Would you expect emergence vaccination to improve outbreak scale / duration if Average duration was 26 day and breakthrough defined 42 days?

• There are any serology / antibody-titers done on vaccinated kids?

• How do 83% of children have herps? How do they define “sporadic herpes “?

• How many children received 2 vaccine doses (prior to outbreak) if policy change in 2018?

• Figure 3-4 bad fits for distribution and regression (clear outlier is holding the graph). Should try a robust fit omit outlier.

• Clearer chronology of varicella public- health policy changes (vaccine introduction, emergence vaccination, free 2 dose)

• Algorithm for management of outbreak

line 53 change administrate to administer

line 72 add each to for outbreak

6. PLOS authors have the option to publish the peer review history of their article (what does this mean?). If published, this will include your full peer review and any attached files.

Reviewer #1: **Yes: **Abdullah Alqarihi

While revising your submission, please upload your figure files to the Preflight Analysis and Conversion Engine (PACE) digital diagnostic tool, https://pacev2.apexcovantage.com/. PACE helps ensure that figures meet PLOS requirements. To use PACE, you must first register as a user. Registration is free. Then, login and navigate to the UPLOAD tab, where you will find detailed instructions on how to use the tool. If you encounter any issues or have any questions when using PACE, please email PLOS at figures@plos.org. Please note that Supporting Information files do not need this step

---

## [Author Response · Author response to Decision Letter 0]

20 May 2022

PONE-D-21-36096

Varicella outbreaks in schools and kindergartens in Shanghai, China from 2011 to 2020

General response: We sincerely thank the editor and reviewer for their valuable feedback that we have used to improve the quality of our manuscript. The comments are laid out below. Our response is given below and changes/additions to the manuscript are using the 'Track Changes' tool in Microsoft Word so that all revisions are clearly visible. 

To ACADEMIC EDITOR：

1. I suggest using a statistical trend analysis (e.g., Mann–Kendall trend) to evaluate the increase or decrease in the number of cases over the study period.

Response: Thank you for the comments. As the editor suggested, we used Mann–Kendall trend test, and the result showed an upward trend in the number of cases over the study period. Please see in lines 125-128.

2. I observed some values of F-statistic and P in different places in the result section. However, the use of different statistical tests to generate these values is missing in the statistical analysis section. Please incorporate.

Response: Thank you for the comments. As the editor suggested, we updated information in the statistical analysis section. Please see in lines 98-106.

3. Please describe the method to calculate the incidence risk. So far, I understand the incidence rate [number of new cases divided by the total person-time at risk] was not calculated and so please replace incidence rate by incidence risk in Figure 2. Moreover, please replace the incidence rate (1/1000) with the incidence risk (per 10000) in the right vertical axis of figure 2.

Response: Thank you for the comments. As the editor suggested, we have described this part in lines 94-97 and replaced Fig 2. 

To Reviewer：

Reviewer 1: Reviewer #1: Wang et al presented the number of varicella outbreaks from 2011 to 2020 in China.

Response: Thank you so much for all the valuable comments. It is very helpful. 

1. In your background would add more information about Varicella symptoms since you talked about it in your discussion.

Response: Thank you for the comments. It is usually a childhood infection, as the majority of the cases occur in people younger than 6 years [1]. The disease manifests as a mild skin rash but complications are possible However, complications such as encephalitis, pneumonitis and secondary bacterial infections may occur, resulting in hospitalization and deaths [2]. Information updated in lines 48-53.

2. The number of varicella outbreaks has not changed significantly in the decade from 2011 to 2020 at least in the abstract, no inclusion of trend data (instead supplied only average data).

Response: Thank you for the comments. We used Mann–Kendall trend test, and the result showed an upward trend in the number of cases over the study period. Information updated in lines 125-128.

3. What do you mean in line 72-73 when you said case info is collected and a report was written? was it done by you or an official source?

Response: Thank you for the comments. A report should be written for each varicella outbreak according to the varicella control and prevention regulation in Shanghai. The report was done by me and my colleagues. We updated the information in lines 73-77.

4. Can you explain what do you mean when you mentioned in line 122 on the second day?

Response: Thank you for the comments. Emergency vaccination took out in 12 outbreaks (28.6%) two days after the report of the secondary varicella case. We are so sorry about the inaccurate translation. The article was language polished by the professional language editing service to improve the writing and to make the revised manuscript easy to follow. We updated the information in lines 139-140.

5. Line 132 it would be nice if you change the title from BV cases to breakthrough Varicella cases.

Response: Thank you for the comments. We have replaced BV cases to breakthrough varicella cases.

6. From lines 148 to 152 it is not clear what are you trying to say. the sentences are not connected.

Response: Thank you for the comments. We are so sorry about the inaccurate translation. The article was language polished by the professional language editing service to improve the writing and to make the revised manuscript easy to follow. We updated the information in lines 162-166.

7. Why does China only give one dose (compared to standard of practice in US of 2 doses)? Does this discrepancy drive outbreak?

Response: Thank you for the comments. Varicella vaccine is not included in the National Immunization Program in China. But two doses of vaccine are free for children in Shanghai since 2018. There may be some reasons such as economic situation of local government. Whether it drive the outbreak need further evaluation.

8. Clarify whether 2nd dose five if patients already received a dose prior to the outbreak

Response: Thank you for the comments. 357 (62.3%) had a history of varicella VarV, in which 345 (96.6%) cases had a history of one dose and 12 (3.4%) cases had two doses of VarV.

9. How did they diagnose/define Varicella? Are there any clinical basis or labs?

Response: Thank you for the comments. There was only clinical diagnosis instead of labs. We intend to do PCR test for varicella diagnosis in some schools in 2022. But it was delayed due to the COVID-19 outbreak in Shanghai recently which caused school closure again.

10. Was there an explanation of statistical analysis of trend?

Response: Thank you for the comments. We used Mann–Kendall trend test and the result showed an upward trend in the number of cases over the study period. Please see in lines 102-106.

11. Would you expect emergence vaccination to improve outbreak scale / duration if Average duration was 26 day and breakthrough defined 42 days?

Response: As we can see from the results emergency vaccination had some effect on the control of varicella outbreak. But it is hard to say whether it is because of the schools carried out emergency vaccination paid more attention on varicella outbreaks and took some other control and prevention measures. Thank you so much for the comments, it is a good thought for further research and varicella policy change.

12. There are any serology / antibody-titers done on vaccinated kids?

Response: Thank you for the comments. There are no such serology / antibody-titers test in the hospitals. All cases are clinical diagnosed. We plan to do PCR test for varicella diagnosis in some schools in 2022. But the it was delayed due to the COVID-19 outbreak in Shanghai recently which caused school closure again. 

13. How do 83% of children have herps? How do they define “sporadic herpes ”?

Response: Thank you for the comments. It was the clinical symptoms of breakthrough varicella cases. 295 cases counted for 82.6% had vesicle. We are so sorry about the inaccurate translation. The article was language polished by the professional language editing service to improve the writing and to make the revised manuscript easy to follow. We updated the information in line 157.

14. How many children received 2 vaccine doses (prior to outbreak) if policy change in 2018?

Response: Thank you for the comments. There was no VZV case received 2 doses of VarV before 2018.

15. Figure 3-4 bad fits for distribution and regression (clear outlier is holding the graph). Should try a robust fit omit outlier.

Response: Thank you for the comments. We deleted Fig 3-4, and described the results in words. 

16. Clearer chronology of varicella public- health policy changes (vaccine introduction, emergence vaccination, free 2 dose) algorithm for management of outbreak

Response: Thank you for the comments. VarV was first used in China in 1998, and the current vaccination procedure is to administer only one dose of VarV for children over 12 months of age. In order to control the outbreak of varicella effectively, Shanghai began to implement emergency vaccination of VarV in schools and kindergartens since 2014. Then free two doses of VarV vaccination strategy implemented in Shanghai since 2018. Please see in lines 56-59.

17. Line 53 change administrate to administer

Response: Thank you for the comments. We have updated it in line 55.

18. Line 72 add each to for outbreak 

Response: Thank you for the comments. We have updated it in line 73.

---

## [Decision Letter · Decision Letter 1]

15 Jun 2022

Varicella outbreaks in schools and kindergartens in Shanghai, China from 2011 to 2020

PONE-D-21-36096R1

Dear Dr. Gao,

We’re pleased to inform you that your manuscript has been judged scientifically suitable for publication and will be formally accepted for publication once it meets all outstanding technical requirements.

Kind regards,

A. K. M. Anisur Rahman, Ph.D.

Academic Editor

PLOS ONE

Additional Editor Comments (optional):

Reviewers' comments:

%REVIEW_QUESTIONS_AND_RESPONSES%

---

## [Editor Report · Acceptance letter]

22 Jun 2022

PONE-D-21-36096R1 

Varicella outbreaks in schools and kindergartens in Shanghai, China from 2011 to 2020 

Dear Dr. Gao:

I'm pleased to inform you that your manuscript has been deemed suitable for publication in PLOS ONE. Congratulations! Your manuscript is now with our production department. 

Kind regards, 

on behalf of

Dr. A. K. M. Anisur Rahman 

Academic Editor

PLOS ONE